# Physical Activity in Women with Endometriosis: Less or More Compared with a Healthy Control?

**DOI:** 10.3390/ijerph20176659

**Published:** 2023-08-26

**Authors:** Maike Katja Sachs, Ioannis Dedes, Samia El-Hadad, Annika Haufe, Dalia Rueff, Alexandra Sabrina Kohl Schwartz, Felix Haeberlin, Stephanie von Orelli, Markus Eberhard, Brigitte Leeners

**Affiliations:** 1Department of Reproductive Endocrinology, University Hospital Zurich, Frauenklinikstrasse 10, 8091 Zurich, Switzerland; samia.el-hadad@usz.ch (S.E.-H.); annika.haufe@usz.ch (A.H.); dalia.rueff@usz.ch (D.R.); brigitte.leeners@usz.ch (B.L.); 2Division of Gynecology and Obstetrics, Inselspital Bern, 3010 Bern, Switzerland; ioannis.dedes@insel.ch; 3Division of Gynecological Endocrinology and Reproductive Medicine, Cantonal Hospital of Lucerne, 6000 Lucerne, Switzerland; 4Department of Gynecology and Obstetrics, Canton Hospital St. Gallen, 9007 St. Gallen, Switzerland; felix.haeberlin@hin.ch; 5Department of Gynecology and Obstetrics, Triemli Hospital Zurich, 8063 Zurich, Switzerland; stephanie.vonorelli@zuerich.ch; 6Department of Gynecology and Obstetrics, Canton Hospital Schaffhausen, 8208 Schaffhausen, Switzerland; markus.eberhard@spitaeler-sh.ch

**Keywords:** endometriosis, physical activity, exercise

## Abstract

Background: Endometriosis, i.e., endometrial-like tissue outside the uterus, is a chronic inflammatory condition affecting physical functioning. However, the specific levels of physical activity (PA) in the context of endometriosis and different disease symptoms remain unclear. Methods: This multi-center, cross-sectional study compared PA levels and influencing factors in endometriosis patients and non-endometriosis patients. Data were collected through questionnaires. Endometriosis was surgically confirmed. A statistical analysis was performed with appropriate tests. Results: The study included 460 women with endometriosis and 460 age-matched women without this condition. The two groups did not differ significantly in terms of age, education level, or stable partnership. Women with endometriosis exhibited lower PA levels, practicing fewer hours of sports weekly and climbing fewer stairs daily compared to the control group. These differences remained significant after controlling for confounding factors. Factors such as endometriosis, current dysmenorrhea, and depression were associated with decreased PA. Conclusions: These findings suggest that women with endometriosis engage in less PA compared to those without this condition. These results highlight the need for interventions to promote increased PA in endometriosis patients and harness the associated health benefits. Further research is warranted to explore the underlying mechanisms and develop tailored exercise therapies for this population.

## 1. Introduction

Endometriosis, a chronic inflammatory condition characterized by the presence of endometrial-like tissue outside the uterus, exerts a significant impact on physical functioning in affected individuals [1]. 

While a decrease in physical function might lead to reduced physical activity (PA) on the one hand, it has also been reported as a helpful measure to reduce endometriosis-related pain [2,3]. Additionally, comorbidities that are associated with endometriosis, including chronic pelvic pain, fatigue, and depression, pose challenges for patients in maintaining regular PA [4]. Understanding the interplay between PA and endometriosis may lead to effective pain management strategies and possible insights into the development of the disease and its symptoms. However, the specific PA levels of individuals with endometriosis have not been studied.

Animal studies have demonstrated that PA has the capacity to impede the growth of endometriosis due to the potential anti-inflammatory effects of myokines secreted by the skeletal muscle [5], suggesting a potential protective role in humans. However, meta-analyses have not conclusively supported these findings in women [2]. 

In the broader context of chronic pain, PA has shown favorable effects in reducing pain severity and improving physical function [6]. However, the effects were mostly of a small-to-moderate magnitude and exhibited variability across studies. Nonetheless, PA has been associated with a reduced risk of dysmenorrhea [2]. A systematic review and meta-analysis focusing on primary dysmenorrhea demonstrated that various types of physical exercise, including aerobic, resistance, and stretching exercises, can effectively alleviate pelvic pain and enhance quality of life [3]. 

Animal model research revealed that PA intervention reduced endometriotic lesions and inflammation and enhanced antioxidant defense [7]. However, human clinical studies on the positive effects of PA intervention for reducing endometriosis-related pain are limited [8] and inconclusive [9,10,11]. Also, in patient-reported outcomes of self-management strategies in the management of endometriosis, forms of PA such as yoga/pilates, stretching, and exercise were rated as being less effective, with a high percentage of adverse events in 34.3% [12].

Several hypotheses have been proposed to explain the potential mechanisms through which physical activity influences endometriosis. One such hypothesis involves myokines, a group of cytokines including IL-6, irisin, and myostatin, that are released by working muscles and contribute to anti-inflammatory, metabolic, and immunological processes [13]. Conflicting results exist regarding the role of IL-6 in endometriosis, as it can elicit a pro-inflammatory response and contribute to lesion growth [14]. It has been postulated that the pro-inflammatory cytokine IL-6 may exert anti-inflammatory effects when secreted by working muscles, and various involved pathways and/or cofactors determine the effect of IL-6 [13].

Physical activity also stimulates the release of β-endorphins, which may modulate pain perception by activating µ-opioid receptors [15]. Nonetheless, dysfunction in endogenous analgesia has been observed in chronic pain patients, indicating the need for tailored exercise therapy.

Another potential action of PA on endometriosis can be postulated on the hormonal axis: PA induces a decrease in circulating sex hormones, especially estrogen levels. As endometriosis is characterized by an estrogen dependence, PA could therefore further promote an anti-inflammatory and antiproliferative effect, as has been investigated in breast cancer patients [16].

While PA has been investigated as an intervention for endometriosis and endometriosis-related pain, this study aimed to answer the question of whether the actual PA levels in women with endometriosis are different from those in women without this condition. Further, this study explored factors that independently contribute to decreased PA in endometriosis and control patients, which could provide valuable insights and potential targets for promoting increased PA and harnessing its associated health benefits.

## 2. Materials and Methods

### 2.1. Study Design

This study was a multi-center, cross-sectional study of an age-matched cohort of women comparing PA and its potential influencing factors between a cohort of endometriosis patients and a cohort of non-endometriosis patients. The main outcome was self-reported PA. 

### 2.2. Data Collection

The recruitment of study participants was conducted at different hospitals in Switzerland, Austria, and Germany: the University Hospital Zurich; the Triemli Hospital Zurich; the district hospitals in Schaffhausen, St. Gallen, Baden, Winterthur, Solothurn, and Walenstadt; the University Hospital Graz; the University Hospital Aachen; the Charitê, Berlin; the Vivantes Humboldt-Klinikum, Berlin; and the Albertinen Hospital, Hamburg. 

Interested patients were provided with written information, a consent form for study participation, a consent form for the verification of their diagnosis with the operation report collected from the treating hospital (if there was an underlying endometriosis diagnosis), a questionnaire, and a return envelope. The recruitment setting for women with endometriosis was either a post-operative follow-up or a regular yearly check-up, while women in the control group were recruited during a yearly check-up or when undergoing mild gynecological interventions (i.e., the laparoscopic removal of a simple ovarian cyst), provided that they met the inclusion criteria of the study. 

### 2.3. Inclusion Criteria

For the disease group, women with intraoperatively and histologically confirmed endometriosis were included, while women in the control group either showed no pelvic pain or had endometriosis excluded via laparoscopy or laparotomy. In order to classify the diagnosis of endometriosis, the revised American Society for Reproductive Medicine (rASRM) classification was used. Women with chronic pelvic pain deriving from diseases other than endometriosis were also excluded. All participating women had to exhibit a sufficient knowledge of the language in which the distributed questionnaire was written and psychological capacity to respond adequately to it. Only questionnaires with ≥80% answers for the main outcome measures were included in this analysis.

The exclusion criteria for both groups were pregnancy and/or a severe pre-existing health condition, i.e., an underlying diagnosis of anorexia, congestive heart disease, Crohn’s disease/colitis ulcerosa, chronic back pain, rheumatic disease, diabetes, osteoporosis, apoplexy, or cancer. Age ±3 years was used as the matching criterion. Following these criteria, of a total of 920 age-matched women, 460 women with endometriosis and 460 women without endometriosis met the inclusion criteria.

### 2.4. Questionnaire and Definitions

The self-administered questionnaire was developed by specialists in the field of endometriosis and psychosomatic medicine from the university hospitals of Berlin and Zurich, incorporating the experience of board members of endometriosis self-help groups. The questionnaire comprises factors related to endometriosis and quality of life. Furthermore, it addresses pre-existing health conditions, demographic information, pregnancy, partnership, lifestyle, and psychological disorders, amongst other factors. Further details on the complete questionnaire are available at ClinicalTrials.gov, (accessed on 1 June 2023) (NCT 02511626). The collected data were entered into an “Access” database after encryption. 

Information on endometriosis-related pain was retrieved using questions from the Brief Pain Inventory and the Pain Disability Index, whereas depression was classified using the Patient Health Questionnaire (PHQ-9) with a score ≥ 10 [17]. The body mass index (BMI) was calculated according to the World Health Organization guidelines, with the formula BMI = kg/m^2^. 

### 2.5. Statistical Analysis

The chi-square test was performed to assess differences with respect to categorial variables, and *t*-tests were applied to test the equality of the group means for continuous variables. A *p*-value of less than 0.05 was considered statistically significant. Subsequently, we performed a multinomial regression analysis, as well as using crude and adjusted odds ratios to evaluate parameters associated with PA. For the multinominal regression analysis, we included all parameters which were significant on the bivariate level (*p* < 0.05). For the statistical tests, IBM SPSS, Version 24, Armonk, NY, USA, was used.

## 3. Results

We enrolled 460 women with endometriosis and 460 age-matched women without endometriosis. The mean age was 37.0 (±6.5) years in the endometriosis group and 37.9 (±8.5) years in the control group (*p* = 0.988) (Table 1). The education level and stable partnership were not different between groups (*p* = 0.355 and *p* = 0.655, Table 1). The patient’s income and family income were higher in the control group (*p* = 0.017 and *p* = 0.033), and parity was also higher in the control group (*p* < 0.002) (Table 1). 

Table 2 reveals the proportions of ASRM stage I/II (mild–moderate endometriosis, N = 162) and ASRM stage III/IV (severe endometriosis, N = 268) women with endometriosis whose stage of disease was indicated on the underlying surgery report. Women with severe endometriosis underwent a significantly higher number of laparoscopic endometriosis surgeries (*p* < 0.001), and the time interval from the initial diagnosis and the last surgery was greater in comparison with the mild–moderate endometriosis group (*p* < 0001 and *p* = 0.038, Table 2).

Current dysmenorrhea was shown to be higher in women with endometriosis, while the occurrence of primary dysmenorrhea in the patient’s history did not differ between the two groups (*p* < 0.001 and *p* = 0.854, Table 3). Hormonal contraceptive use was more common in the control group (*p* < 0.010), whereas women with endometriosis had depression more often than women without endometriosis (*p* < 0.001, Table 3). 

The comparison of the endometriosis subgroups (mild–moderate, ASRM stage I/II vs. severe, ASRM stage III/IV endometriosis) revealed that current dysmenorrhea, primary dysmenorrhea, endometriosis-related chronic pain, hours of endometriosis-related pain (per day), hormonal contraceptive use, fatigue, and depression were not different between the two groups (*p* = 0.489, *p* = 0.854, *p* = 0.486, *p* = 0.327, *p* = 0.420, *p* = 0.536 and *p* = 0.172, respectively, Table 3).

Physical activity, described according to weekly hours of sports and number of stairs climbed daily, differed between women with and without endometriosis. Women with endometriosis practiced less hours of sports weekly (*p* < 0.001) and climbed less stairs daily (*p* < 0.038, Table 4). Here, statistical significance remained in the multinominal logistic regression analysis, controlling for the patient’s income, family income, parity, current dysmenorrhea, hormonal contraceptive use, and depression for the parameter “hours of sports weekly” but not for the number of stairs climbed daily (Table 4). The stratification based on the ASRM stages did not reveal differences in physical activity (Table 4). Other parameters of health (smoking, drugs, BMI) were distributed equally between women with and without endometriosis and between the two subgroups of endometriosis (Table 4). 

To further investigate potential factors affecting lower sports activity (defined as <3 h per week), crude and adjusted odds ratios (cOR and aOR) were calculated for the total study population (Table 5), the group of endometriosis patients (Table 6), and the group of control patients without endometriosis (Table 7). 

When analyzing the whole patient population, endometriosis (1.89 [1.42; 2.52], current dysmenorrhea (1.85 [1.23; 2.22] and depression (1.98 [1.26; 3.12] showed a significant cOR, while endometriosis (1.76 [1.19–2.62]) and higher parity (1.59 [1.07–2.36]) showed significant aORs (Table 5). Differences in patient income or family income did not affect sports activity (Table 5).

The factors affecting sports activity in the sub-cohort of control women did not reveal a significant cOR, but non-nulliparous status had an aOR of 1.86 [1.10–3.16] for lower sports activity (Table 6). 

In the sub-cohort of women with endometriosis, we revealed that endometriosis-related chronic pain had a cOR of 1.86 [1.19–2.90] for lower sports activity, whereas the aOR of endometriosis-related chronic pain was not significant (Table 7). Of note, a shorter time interval to the latest laparoscopy did not have a significant influence on PA (Table 7).

## 4. Discussion

The results of our study provide compelling evidence that women with endometriosis engage in less PA than their counterparts without this disease. We found that women with endometriosis practiced fewer hours of sports weekly and climbed fewer stairs daily compared to women without endometriosis. Importantly, the differences in weekly hours of sports remained significant, even after adjusting for confounding variables such as depression, income, parity, current dysmenorrhea, and hormonal contraceptive use. This suggests that endometriosis itself is an independent factor leading to decreased PA levels. 

To our best knowledge, this is the first study to report on PA in women with endometriosis in comparison with a healthy control, while others have mainly addressed the influence of PA on pain reduction and well-being [5,8,9,11,12,19]. 

Lower PA levels in women with endometriosis may contribute to a sedentary lifestyle, potentially exacerbating the already burdensome symptoms associated with this condition and potential long-term health implications [20]. The observed reduced PA among women with endometriosis is concerning, as regular PA is known to have numerous health benefits, including mental well-being and reduced chronic disease risk [21,22,23]. Of particular concern is lower PA in regard to emerging evidence of an increased cardiovascular risk associated with an atherogenic lipid profile [24] and atherosclerosis [25] in endometriosis, as it is established that PA improves cardiovascular fitness. 

The decreased levels of PA in women with endometriosis can be attributed to various factors, with dysmenorrhea being the most prominent symptom. In our study, the presence of dysmenorrhea was higher in women with endometriosis compared to the control group, as expected. Dysmenorrhea was identified as a factor contributing to lower PA levels across the entire study population. However, after adjusting for confounding variables, the direct effect of dysmenorrhea on PA was not sustained. Nevertheless, the previous literature has already established an association between lower PA and dysmenorrhea.

Chronic pain is another significant factor that can impede PA, leading to an avoidance behavior. Our findings demonstrate that endometriosis-related chronic pain was associated with a higher likelihood of reduced PA within the group of women with endometriosis, as indicated by the lower crude odds ratio. This supports the notion that chronic pain experienced by individuals with endometriosis plays a role in limiting their engagement in PA.

It is important to note that dysmenorrhea and chronic pain are interconnected symptoms in endometriosis, and their impacts on PA levels may be intertwined. While dysmenorrhea alone did not maintain a direct effect on PA after considering the confounders, it has been consistently described in the literature as a factor contributing to decreased PA. Together with chronic pain, these symptoms pose significant challenges for women with endometriosis in maintaining regular PA levels. 

Understanding and addressing these factors are crucial for developing tailored strategies to promote PA and mitigate the impact of endometriosis on women’s overall well-being. 

The promotion of PA for all patients with endometriosis needs to be approached cautiously, considering the limited evidence regarding its effectiveness. The current literature [9,11] lacks controlled and randomized studies that can definitively determine the extent to which PA prevents the progression and alleviates the symptoms of endometriosis. As a result, the specific impact of PA on this condition remains unclear. Armour et al. reported adverse events with exercise which may be attributed to inadequate PA (i.e., vigorous exercise may worsen pain symptoms) [12]. Similarly, it has been postulated that exercise therapy should be adapted on an individual basis in order to circumvent a flare of symptoms [15]. A Cochrane review underlined the low quality of evidence for PA’s effect on chronic pain in general due to inconsistent outcomes with a mostly minimal effect [6]. In contrast, the possible positive effects of PA on pain cannot be denied [5,6,10]; for example, yoga practice was associated with a reduction in pain in women with endometriosis [8].

Furthermore, PA has been shown to reduce anxiety and promote positive stress-coping strategies [26]. With regard to the higher incidence of depression according to the PHQ-9 in our endometriosis cohort and the well-attested higher prevalence of fatigue in endometriosis [27], targeted interventions to promote PA seem reasonable. It is noteworthy that endometriosis remained a significant independent factor for decreased PA even after correcting for depression in this study. This suggests that endometriosis exerts its influence on PA levels through mechanisms beyond the presence of depression alone. A limitation of the PHQ-9 for depression and the questions regarding PA utilized in this study is its self-reported nature, whereas objective measures may result in different estimates of PA. It has been postulated that depressive patients may over- or under-report their PA, meaning that, ideally, we require objective measures for PA [4]. 

Hormonal contraceptive use was more common in the control group, which may have also contributed to the differences in PA levels. Therefore, PA was corrected for hormonal contraceptive use in this study, and women with endometriosis still had lower PA. Hormonal contraceptives are frequently prescribed to manage symptoms of endometriosis, such as pain and heavy menstrual bleeding. However, studies have suggested that hormonal contraceptives can affect energy levels and motivation for PA, and further investigation is warranted to understand the impact of hormonal contraceptives on PA levels in women with endometriosis [28,29,30].

The severity of endometriosis, defined according to the rASRM stage, was not associated with differences in PA, nor did we observe any other differences in parameters between the two groups. Epidemiological and genetical evidence has consistently shown a lack of correlation between rASRM stage and pelvic pain severity [31,32]. This poor correlation between the symptoms and stage of endometriosis is consistent across different classification systems, including the #ENZIAN [33] system, which is known to enable better discrimination for deep endometriosis [34] but was not available in this study. 

Even though income was higher in the control group, it was not associated with a difference in PA. Observing the overall population, independent of endometriosis status, parity was found to be an independent risk factor for lower PA, meaning that women with at least one child had relevant odds for lower PA. This phenomenon has previously been described in the literature [35].

The limitations of this study include the self-administered nature of the questionnaire, which is subjective and may be influenced by a recall bias. Medication, except for hormonal contraceptives, was not known (i.e., pain killers), nor did we know if the participant was enrolled in a program promoting PA. We cannot exclude the possibility that the patients in the endometriosis groups had concomitant adenomyosis, and even more importantly, we can assume that an unknown number of women belonging to the control group had underlying, non-diagnosed adenomyosis. As adenomyosis is known to cause endometriosis-like symptoms, this may diminish the difference between our two groups and weaken the statistical differences we reported within this study. Furthermore, the underlying study design did not allow us to evaluate casual effects. Differences between the groups might have been underestimated, as we cannot exclude the possibility that women in the control group had undiagnosed endometriosis. 

In light of the challenges faced by women with endometriosis in engaging in PA, it is imperative to explore strategies that may facilitate their participation and mitigate pain-related barriers. One promising avenue is the implementation of tailored exercise programs that consider the unique pain profiles of women with endometriosis. Such programs could encompass a comprehensive approach involving a blend of aerobic, resistance, and flexibility exercises. By designing exercises that specifically cater to the pain thresholds and limitations experienced by these individuals, it may be possible to alleviate their discomfort.

Furthermore, elucidating the underlying mechanisms linking endometriosis-related pain and physical activity should remain a primary focus of future research endeavors. Investigating the intricate interplay between pain perception pathways, inflammatory processes, and neurophysiological responses could help to uncover novel therapeutic targets. 

To this end, interdisciplinary collaborations between gynecologists, pain specialists, physiotherapists, and neuroscientists hold immense potential for unraveling the intricate relationship between pain/inflammation and physical activity in the context of endometriosis. By systematically investigating these interventions through robust methodologies, the academic community could pave the way for evidence-based strategies that empower women with endometriosis to leverage physical activity as a valuable tool in their pain management regimens.

## 5. Conclusions

Our study provides evidence that women with endometriosis exhibit reduced levels of PA compared to age-matched women without this condition. Increased pain and pain sensitivity, fear and avoidance behaviors, fatigue, and limited social support may limit engagement in PA among these women. Addressing these comorbidities and tailoring interventions to account for mental health considerations are essential for promoting PA and overall well-being in women with endometriosis.

Future research should further explore the underlying mechanisms and potential interventions to address this issue.

## Figures and Tables

**Table 1 ijerph-20-06659-t001:** Socio-demographic information.

Parameter	Endometriosis	Control	*p*-Value
**Numbers**	N = 460	N = 460	
**Age (years)**	37.0 (±6.5)	37.9 (±8.5)	0.988
**Education**	N = 455	N = 455	0.355
Primary School	21 (4.6%)	13 (2.8%)	
Secondary School	44 (9.7%)	45 (9.9%)	
Apprenticeship	178 (39.1%)	203 (44.4%)	
University	173 (38.0%)	165 (36.1%)	
Others	38 (8.3%)	28 (6.2%)	
None	1 (0.2%)	3 (0.7%)	
**Patients’ Income**	N = 437	N = 440	**0.017**
No Income	46 (10.5%)	17 (4.4%)	
<3000 CHF	92 (21.1%)	14 (3.7%)	
<4000 CHF	73 (16.7%)	13 (3.4%)	
<5000 CHF	93 (21.3%)	34 (8.9%)	
<6000 CHF	60 (13.7%)	42 (11%)	
>6000 CHF	73 (16.7%)	263 (68.7%)	
**Family Income**	N = 380	N = 383	**0.033**
No Income	21 (5.5%)	17 (4.4%)	
<3000 CHF	18 (4.7%)	14 (3.7%)	
<4000 CHF	16 (4.2%)	13 (3.4%)	
<5000 CHF	48 (12.6%)	34 (8.9%)	
<6000 CHF	43 (11.3%)	42 (11.0%)	
>6000 CHF	234 (61.6%)	263 (68.7%)	
**Stable partnership**	N = 459	N = 458	0.655
Yes	380 (82.8%)	374 (81.7%)	
**Parity**	N = 421	N = 439	**<0.001**
**P0**	294 (69.8%)	222 (50.6%)	
**P1**	75 (17.8%)	70 (15.9%)	
**P2**	44 (10.5%)	107 (24.4%)	
**Parity ≥ 3**	8 (1.9%)	40 (9.1%)	

Data presented as n (%) or mean (±SD) using chi-square and independent *t*-tests. SD = standard deviation; CHF = Swiss franc. The bold values indicate statistical significance. P0 = nullipara, P1 = one child born, P2 = two children born, Parity ≥ 3 = three or more children born.

**Table 2 ijerph-20-06659-t002:** Endometriosis characteristics.

Parameter	ASRM Stage I/II (N = 162)	ASRM Stage III/IV (N = 268)	*p*-Value
**Number of endometriosis surgeries**	1.4 (±0.70)	2.0 (±1.3)	**<0.001**
**Time from initial diagnosis (months)**	37.7 (±41.2)	62.0 (±34.7)	**<0.001**
**Time from last surgery (months)**	26.9 (±29.0)	34.7 (±35.4)	**0.038**

Data presented as N (%) or mean (±SD) using chi-square and independent *t*-tests. SD = standard deviation. The bold values indicate statistical significance. For ASRM (= American Society for Reproductive Medicine) stages I–IV, see [18].

**Table 3 ijerph-20-06659-t003:** Menstruation, pain, and emotional health.

Parameter	Endometriosis	ASRM Stages I/II	ASRM Stages III/IV	Control	Endometriosis vs. Control (*p*-Value)	Endometriosis ASRM Stages I/II vs. III/IV (*p*-Value)
**Current dysmenorrhea**	N = 447	N = 177	N = 268	N = 435	**<0.001**	0.489
Yes	284 (63.5%)	116 (65.6%)	167 (62.3%)	120 (27.6%)		
**Primary dysmenorrhea**	N = 394	N = 157	N = 236	N = 231	0.739	0.854
Yes	178 (45.2%)	72 (45.9%)	106 (44.9%)	101 (43.7%)		
**Endometriosis-related chronic pain**	N = 460	N = 181	N = 277	N/A		0.486
Yes	218 (47.4%)	89 (49.2%)	127 (45.8%)	N/A		
**Hours of endometrio-sis-related pain (per day)**	N = 241	N = 100	N = 139	N/A		0.327
<4 h	80 (33.2%)	37 (37.0%)	43 (30.9%)	N/A		
≥ 4 h	161 (66.8%)	63 (63.0%)	96 (69.1%)	N/A		
**Hormonal contracep-tive use**	N = 442	N = 170	N = 270	N = 428	**0.010**	0.420
Yes	158 (35.7%)	65 (38.2%)	93 (34.4%)	190 (44.4%)		
**Fatigue**	N = 253	N = 105	N = 131	N/A		0.536
Yes	221 (87.4%)	90 (85.7%)	129 (88.4%)			
**Depression**	N = 431	N = 172	N = 257	N = 405	**<0.001**	0.172
Yes	113 (26.2%)	51 (29.7%)	61 (23.7%)	33 (8.1%)		

Data presented as N (%) or mean (±SD) using chi-square and independent *t*-tests. SD = standard deviation. The bold values indicates statistical significance. Depression = Patient Health Questionnaire (PHQ-9) score ≥ 10.

**Table 4 ijerph-20-06659-t004:** Physical activity and health behavior.

Parameter	Endometriosis	Endometriosis ASRM Stages I/II	Endometriosis ASRM Stages III/IV	Control	Endometriosis vs. Control *p*-Value	Endometriosis ASRM Stages I/II vs. III/IV *p*-Value
**Hours of sports weekly**	N = 427	N = 161	N = 264	N = 438	**<0.001 ***	0.136
**None**	126 (29.5%)	41 (25.5%)	84 (31.8%)	90 (20.5%)		
**<3**	189 (44.3%)	76 (47.2%)	112 (42.4%)	172 (39.3%)		
**<6**	99 (23.2%)	42 (26.1%)	57 (21.6%)	151 (34.5%)		
**>6**	13 (3.0%)	2 (1.2%)	11 (4.2%)	25 (5.7%)		
**Number of stairs daily**	N = 428	N = 162	N = 264	N = 439	**0.038 ****	0.584
None	13 (3.0%)	7 (4.3%)	6 (2.3%)	14 (3.2%)		
<10	147 (34.3%)	57 (35.2%)	90 (34.1%)	116 (26.4%)		
10–50	224 (52.3%)	80 (49.4%)	142 (53.8%)	244 (55.6%)		
>50	44 (10.3%)	18 (11.1%)	26 (9.8%)	65 (14.8%)		
**Smoking**	N = 428	N = 162	N = 264	N = 436	0.161	0.721
Yes	96 (22.4%)	38 (23.5%)	58 (22.0%)	81 (18.6%)		
**Drug**	N = 428	N = 161	N = 265	N = 436	0.089	0.305
Yes	16 (3.7%)	8 (5.0%)	8 (3%)	8 (1.8%)		
**BMI (kg/m^2^)**	N = 458 22.9 (±4.32)	N = 181 22.7 (±4.55)	N = 275 22.9 (±4.16)	N = 457 22.8 (±4.27)	0.831	0.977

* Statistical significance remained after controlling for the patient’s income, family income, parity, current dysmenorrhea, hormonal contraceptive use, and depression in the multinominal logistic regression analysis. ** Statistical significance did NOT remain after controlling for the patient’s income, family income, parity, current dysmenorrhea, hormonal contraceptive use, and depression in the multinominal logistic regression analysis. Data presented as N (%) or mean (±SD) using chi-square and independent *t*-tests. SD = standard deviation. The bold values indicate statistical significance. BMI = body mass index.

**Table 5 ijerph-20-06659-t005:** Crude odds ratio and adjusted odds ratio from the logistic regression on factors affecting lower sports activity (<3 h per week) in the total population of endometriosis and control patients.

Variables	Grouping of Variables	Crude OR		Adjusted OR	
		OR	95% CI	OR	95% CI
**Endometriosis**	Yes	**1.89 ***	**1.42–2.52**	**1.76 ***	**1.19–2.62**
**Current dysmenorrhea**	Yes	**1.65 ***	**1.23–2.22**	**1.46**	**0.99–2.16**
**Depression**	Yes	**1.98 ***	**1.26–3.12**	**1.29**	**0.74–2.26**
**Patient’s** **Income**	No Income	1.0	-	1.0	-
	<3000 CHF	0.82	0.51–1.32	0.62	0.34–1.11
	<4000 CHF	0.90	0.52–1.55	0.58	0.28–1.15
	<5000 CHF	0.77	0.47–1.27	0.65	0.35–1.21
	<6000 CHF	0.89	0.52–1.53	0.62	0.32–1.22
	>6000 CHF	0.82	0.49–1.39	0.53	0.27–1.04
**Family income**	No Income	1.0	-	1.0	-
	<3000 CHF	0.69	0.23–2.05	0.62	0.17–1.24
	<4000 CHF	1.03	0.36–2.93	1.22	0.36–4.21
	<5000 CHF	0.75	0.32–1.77	0.54	0.19–1.51
	<6000 CHF	1.76	0.78–4.0	1.59	0.60–4.19
	>6000 CHF	1.01	0.49–2.05	0.84	0.35–1.99
**Parity**	P0	1.0	-	1.0	-
	≥ P1	0.90	0.66–1.21	**1.59 ***	**1.07–2.36**

* *p* < 0.05; OR = odds ratio; CI = confidence interval; depression = Patient Health Questionnaire (PHQ-9) Score ≥ 10. CHF = Swiss franc; P0 = nulliparous; P1 = primiparous. The bold values indicate the statistical significance.

**Table 6 ijerph-20-06659-t006:** Crude odds ratio and adjusted odds ratio from the logistic regression on factors affecting lower sports activity (<3 h per week) in the total population of control patients.

Variables	Grouping of Variables	Crude OR		Adjusted OR	
		OR	95% CI	OR	95% CI
**Current dysmenorrhea**	Yes	1.4	0.89–2.17	1.53	0.86–2.71
**Depression**	Yes	1.77	0.76–4.13	1.14	4.23–3.06
**Patient’s** **Income**	No Income	1.0	-	1.0	-
	<3000 CHF	0.83	0.45–1.50	0.53	0.25–1.11
	<4000 CHF	0.99	0.46–2.13	0.54	0.19–1.52
	<5000 CHF	0.75	0.39–1.43	0.61	0.27–1.39
	<6000 CHF	0.51	0.25–1.03	0.29	0.12–0.70
	>6000 CHF	0.63	0.31–1.28	0.31	0.13–0.76
**Family income**	No Income	1.0	-	1.0	-
	<3000 CHF	1.02	0.23–4.6	0.70	0.12–4.17
	<4000 CHF	0.64	0.14–2.91	0.99	0.15–6.5
	<5000 CHF	0.82	0.25–2.71	0.65	0.14–2.97
	<6000 CHF	1.36	0.43–4.30	1.36	0.29–6.30
	>6000 CHF	0.89	0.33–2.43	0.83	0.22–3.16
**Parity**	P0	1.0	-	1.0	-
	≥ P1	1.35	0.91–2.00	**1.86 ***	**1.10–3.16**

* *p* < 0.05; OR = odds ratio; CI = confidence interval; depression = Patient Health Questionnaire (PHQ-9) Score ≥ 10. CHF = Swiss franc; P0 = nulliparous; P1 = primiparous. The bold values indicate the statistical significance.

**Table 7 ijerph-20-06659-t007:** Crude odds ratio and adjusted odds ratio from the logistic regression on factors affecting lower sports activity (<3 h per week) in the total population of endometriosis patients.

Variables	Grouping of Variables	Crude OR		Adjusted OR	
		OR	95% CI	OR	95% CI
**Current dysmenorrhea**	Yes	1.35	0.86–2.10	1.39	0.52–3.71
**Endometriosis-related chronic pain**	Yes	**1.86 ***	**1.19–2.90**	2.92	0.51–16.74
**Hours of endometriosis-related pain (per day)**					
**<4 h**		1.0	-	1.0	-
**≥4 h**		1.36	0.65–2.84	0.97	0.38-2.46
**ASRM Stage III/IV endometriosis**	Yes	1.08	0.67–1.69	1.52	0.64–3.65
**Time interval to laparoscopy**					
**<4 months**		1.0	-	1.0	-
**4–12 months**		0.20	0.79–3.11	0.33	0.06–2.60
**13–24 months**		0.67	0.53–2.67	0.34	0.06–2.65
**>24 months**		0.65	0.53–2.80	0.34	0.43–11.57
**>36 months**		**0.01**	1.22–4.64	0.20	0.63–9.28
**Depression**	Yes	1.6	0.91–2.80	1.11	0.42–2.97
**Fatigue**	Yes	1.5	0.53–4.83	1.07	0.26–4.38
**Patient’s Income**	No Income	1.0	-	1.0	-
	<3000 CHF	0.94	0.38–2.34	0.30	0.07–1.27
	<4000 CHF	1.42	0.57–3.54	0.53	0.13–2.11
	<5000 CHF	1.19	0.49–2.89	0.68	0.16–2.88
	<6000 CHF	2.40	0.97–5.96	0.96	0.23–4.03
	>6000 CHF	1.63	0.67–4.00	0.36	0.07–1.96
**Family income**	No Income	1.0	-	1.0	-
	<3000 CHF	0.43	0.07–2.54	0.38	0.03–5.02
	<4000 CHF	1.60	0.37–6.96	1.03	0.11–9.29
	<5000 CHF	0.67	0.18–2.40	0.13	0.01–1.77
	<6000 CHF	2.23	0.68–7.32	1.15	0.19–6.94
	>6000 CHF	1.01	0.35–2.89	0.36	0.06–2.04
**Parity**	P0	1.0	-	1.0	-
	≥ P1	1.16	0.71–1.89	1.57	0.56–4.52

* *p* < 0.05; OR = odds ratio; CI = confidence interval; depression = Patient Health Questionnaire (PHQ-9) Score ≥ 10. CHF = Swiss franc; P0 = nulliparous; P1 = primiparous. The bold values indicate the statistical significance.

## Data Availability

The data will be made available upon reasonable request.

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
