# Peer review of "Physical Activity in Women with Endometriosis: Less or More Compared with a Healthy Control?"

_ijerph, 2023, doi:10.3390/ijerph20176659_

Round 1

Reviewer 1 Report

Reviewer comments and suggestions

The authors in this study investigated the specific levels of physical activity (PA) in the context of specific endometriosis. The study methods included a multi-center, cross-sectional study comparing PA levels and influencing factors in endometriosis patients and non-endometriosis patients. The study recruited 460 women with endometriosis and 460 age-matched women without the condition. Women with endometriosis exhibited lower PA levels, practicing fewer hours of sports weekly and climbing fewer stairs daily compared to the control group. These differences remained significant after controlling for confounding factors. Hence the study concluded that women with endometriosis engage in less PA compared to those without the condition. 

Overall, the manuscript was well written. However, a few concerns/comments needed to be explained/modified. 

  1. Line 37 The reference cited was not following MDPI journals, please modify the manuscript.
  2. Line 39 is the correct word of using resource here.
  3. Line 45 I think it needed to explain the different comorbidities here only
  4. Line 46 Is there was a mechanism the study suggested if yes please mention.
  5. Line 67-68 Please explore the studies here and why it shows the conflicting results
  6. Line 78 I think the conclusion is not needed to add up here, the authors could modify
  7. Line 119 -123 How many patients the authors approached and how many gave the consent? All information should be needed to write here in the paper.
  8. Comments for Table 1(p0,1,2,3) It should be explained in the legend part
  9. Comments for Table 2 Stage should be defined in the legend part
  10. Line 183-187 Please avoid long sentences.
  11. Line 234-237 These lines do not need to start as it was already mentioned previously
  12.  Comments for discussion section “It would be nice if the authors could add a table or figure for the respective text mentioned in the paragraph”
  13. Line 307-309 Please add up more references here seems there would be more
  14. Check the reference numbers 14,20,25,35 journal style should be abbreviated. 

Author Response

Dear Reviewer 1, we thank you for your thorough revision and suggestions for improvement of this manuscript.

  • Line 37 The reference cited was not following MDPI journals, please modify the manuscript.

 We modified accordingly. Thank you.

  • Line 39 is the correct word of using resource here.

 If I understood correctly you would like us to choose a more suitable wording. Accordingly we replaced “recource” with “measure”:

“While a decrease in physical functioning might lead to reduced physical activity (PA) on the one hand, it has also been reported as a helpful measure to reduce endometriosis-related pain2,3

  • Line 45 I think it needed to explain the different comorbidities here only

 Thank you for mentioning this. We decided to remove the open term “different comorbidities“:

“However, the specific PA levels in individuals with endometriosis have not been studied.”

  • Line 46 Is there was a mechanism the study suggested if yes please mention.

 We thank the reviewer for her/his suggestion to add more details to this part. We improved the mentioned sentence:

“Animal studies have demonstrated that PA has the capacity to impede the growth of endometriosis due to potential anti-inflammatory effects of myokines secreted by skeletal muscle5, suggesting a potential protective role in humans.”

  • Line 67-68 Please explore the studies here and why it shows the conflicting results

 We do agree that a more thorough explanation is interesting for the reader and added the following sentence:

“It has been postulated that the pro-inflammatory cytokine IL-6 may exert anti-inflammatory effects when secreted by the working muscle and various involved pathways and/or cofactors determine the effect of IL-613.”

  • Line 78 I think the conclusion is not needed to add up here, the authors could modify

 We thank the reviewer for pointing this out and deleted the conclusion accordingly.

  • Line 119 -123 How many patients the authors approached and how many gave the consent? All information should be needed to write here in the paper.

 Unfortunately we cannot trace back how many of the approached patients gave consent, but for sure only patients who gave consent were included into the study.

  • Comments for Table 1(p0,1,2,3) It should be explained in the legend part

We thank the reviewer for her/his suggestion and added the following explanation to the legend of Table 1:

“P0=nullipara, P1=one child born, P2= two children born, Parity 3=three or more children born”

  • Comments for Table 2 Stage should be defined in the legend part

 Dear reviewer, we thank you for this suggestion. As the definition contains a large table, we added a reference to the legend part:

“ASRM (=American Society for Reproductive Medicine) Stages I-IV see prior publication18

  • Line 183-187 Please avoid long sentences.

 We thank the reviewer for contributing to a better readability of the manuscript and divided the mentioned sentence into two parts:

“Women with endometriosis practiced less hours of sports weekly (p<0.001) and climbed less stairs daily (p<0.038, Table 4). Here, statistical significance remained on multinominal logistic regression analysis, controlling for the patient`s income, family income, parity, current dysmenorrhea, hormonal contraceptive use and depression for the parameter “hours of sports weekly”, but not for the number of climbed stairs daily (Table 4).”

  • Line 234-237 These lines do not need to start as it was already mentioned previously

 According to your valuable suggestion we removed the sentences from lines 234-237:

«The impact of endometriosis on physical activity (PA) levels has been a subject of growing interest and research. Our study aimed to investigate whether women with endometriosis exhibit reduced PA compared to age-matched women without endometriosis as PA may help to alleviate disease symptoms on the one hand and interfere with PA because of disease symptoms such as pain, fatigue and depression.»

  • Comments for discussion section “It would be nice if the authors could add a table or figure for the respective text mentioned in the paragraph”

 Dear reviewer, we do agree that this would be a nice addition but we did not come to an idea of a table/figure that would substantially add to the discussion section.

  • Line 307-309 Please add up more references here seems there would be more

 Dear reviewer, we added the two following references:

(29)     Gillen, M. M.; Rosenbaum, D. L.; Winter, V. R.; Bloomer, S. A. Hormonal Contraceptive Use and Women’s Well-Being: Links with Body Image, Eating Behavior, and Sleep. Psychology, Health & Medicine 2023, 1–12. https://doi.org/10.1080/13548506.2023.2216468.

(30)     Kristjánsdóttir, J.; Olsson, G. I.; Sundelin, C.; Naessen, T. Self-Reported Health in Adolescent Girls Varies According to the Season and Its Relation to Medication and Hormonal Contraception – A Descriptive Study. The European Journal of Contraception & Reproductive Health Care 2013, 18 (5), 343–354. https://doi.org/10.3109/13625187.2013.821107.

  • Check the reference numbers 14,20,25,35 journal style should be abbreviated. 

 Dear reviewer, we hope that the problem has been solved now due to the updated citation style.

Reviewer 2 Report

Congratulations to this interesting manuscript.

This multi-center, cross-sectional study compared physical activity levels and influencing factors in endometriosis patients and non-endometriosis patients. The topic is extremely relevant in the field of endometriosis, this being the first study to report on physical activity in women with endometriosis in comparison with a healthy control group. The relevance of the study is increased being carried out in a multi-center setting. It is the first study to report on physical activity in women with endometriosis in comparison with a healthy control group and not only focusing on the role of physical activity on endometriosis symptoms.

The references are appropriate.

Perhaps, you can also address the role of adenomyosis in this context in the discussion part. Especially in the control group with no evidence of endometriosis, adenomyosis can represent a confounding factor.  

Author Response

Dear Reviewer 2, the authors thank you for your revision and suggestions for improvement of this manuscript.

According to your suggestion we addressed the role of adenomyosis in the discussion part (see limitations):

We cannot exclude that not only patients of the endometriosis groups have concomitant adenomyosis, but even more importantly we assume that an unknown number of women belonging to the control group have underlying non-diagnosed adenomyosis. As adenomyosis is known to be accountable for endometriosis-like symptoms this may diminish the difference between our two groups and weaken the statistical differences we report within this study.

Reviewer 3 Report

Based on your research, propose some solution that could reduce the sensitivity to pain and fatigue that prevent these women from participating in physical activity. On the basis of your proposal, you can build the foundations of further research that would socially support the involvement of these women in PA.

Author Response

Dear Reviewer 3, we are very thankful for your revision.

Based on your research, propose some solution that could reduce the sensitivity to pain and fatigue that prevent these women from participating in physical activity. On the basis of your proposal, you can build the foundations of further research that would socially support the involvement of these women in PA.

 We thank the reviewer for this suggestions and implemented the following text into our discussion:

“In light of the challenges faced by women with endometriosis in engaging in PA, it becomes imperative to explore strategies that may facilitate their participation and mitigate pain-related hindrances. 

One promising avenue is the implementation of tailored exercise programs that consider the unique pain profiles of women with endometriosis. Such programs could encompass a comprehensive approach involving a blend of aerobic, resistance, and flexibility exercises.

By designing exercises that specifically cater to the pain thresholds and limitations experienced by these individuals, it may be possible to alleviate discomfort.

Furthermore, elucidating the underlying mechanisms linking endometriosis-related pain and physical activity should remain a primary focus of future research endeavors. Investigating the intricate interplay between pain perception pathways, inflammatory processes, and neurophysiological responses could uncover novel therapeutic targets.

To this end, interdisciplinary collaborations between gynecologists, pain specialists, physiotherapists and neuroscientists hold immense potential for unraveling the intricate relationship between pain/inflammation and physical activity in the context of endometriosis. By systematically investigating these interventions through robust methodologies, the academic community can pave the way for evidence-based strategies that empower women with endometriosis to leverage physical activity as a valuable tool in their pain management regimen.”